# Effects of Green Tea Marinade in the Bioaccessibility of Tonalide and Benzophenone 3 in Cooked European Seabass

**DOI:** 10.3390/molecules27154873

**Published:** 2022-07-29

**Authors:** Sara C. Cunha, Juliana R. Gadelha, Flávia Mello, Isa Marmelo, António Marques, José O. Fernandes

**Affiliations:** 1LAQV/REQUIMTE, Laboratório de Bromatologia e Hidrologia, Faculdade de Farmácia, Universidade do Porto, Rua de Jorge Viterbo Ferreira 228, 4050-313 Porto, Portugal; jgadelha@ua.pt (J.R.G.); flavia.bioufrj@gmail.com (F.M.); josefer@ff.up.pt (J.O.F.); 2IPMA, Portuguese Institute for the Sea and Atmosphere, I.P., Division of Aquaculture, Upgrading and Bioprospection (DivAV), Av. Doutor Alfredo Magalhães Ramalho 6, 1495-165 Lisboa, Portugal; isa.marmelo@ipma.pt (I.M.); amarques@ipma.pt (A.M.); 3CIIMAR, Interdisciplinary Centre of Marine and Environmental Research, University of Porto, Rua dos Bragas 289, 4050-123 Porto, Portugal; 4UCIBIO-REQUIMTE, Applied Molecular Biosciences Unit, Department of Chemistry, NOVA School of Science and Technology, NOVA University of Lisbon, 2829-516 Caparica, Portugal

**Keywords:** roasting, frying, green tea marinade, personal care products, GC-MS, fish

## Abstract

Fish consumption is an indicator of human exposure to personal care products (PCP) such as tonalide (AHTN) and benzophenone 3 (BP3). Although most fish consumed is cooked, the impact of cooking procedures on PCP levels is difficult to evaluate. Hence, the aim of this work was to provide thorough information on the stability and bioaccessibility of AHTN and BP3 upon cooking and in vitro digestion. A green tea (*Camellia sinensis*) marinade, rich in polyphenol, was used as mitigating strategy to reduce these contaminants. Roasting and frying reduced AHTN and BP3 levels in European seabass (*Dicentrarchus labrax*) spiked samples. Additionally, the green tea marinade promoted a reduction of up to 47% AHTN and 35% BP3. Bioaccessibility of AHTN was higher (up to 45%), and increased with the use of green tea marinades. BP3 showed a bioaccessibility below 19% in all cooked samples. Overall, a decrease in PCP levels was observed after cooking; this decrease was even more pronounced when marination was previously used. However, this decrease is cancelled out by the fact that the bioaccessible fraction of the contaminants increases in an inverse way; therefore, none of these processes can be considered a mitigating alternative.

## 1. Introduction

Food safety has emerged in recent decades as a global awareness of public health and international trade implications. Among dozens of potentially hazardous contaminants, particular concern, in seafood, comes from the presence of personal care contaminants, such as UV-filters (benzophenones) and musk fragrances (tonalide, galoxilide, cashmeran) [1]. These contaminants have recently been associated with several health disorders, including carcinogenic and reproductive disorders [2,3]. Benzophenone-3 is one of the most widely used UV-filters, and has been available as a sunscreen agent for over 40 years [3]. Due to its release into the aquatic environment, this organic compound has been widely found in aquatic biota, including fish [4].

Among polycyclic musks, tonalide (AHTN) has been repeatedly detected in aquatic biota, such as macroalgae [4], bivalves (clams and mussels) [5], and numerous fish species [2], with levels reaching 349 μg/kg wet weight (ww) [6]. 

Several factors affect contamination levels of UV-filters and polycyclic musks in seafood, including the chemical properties of the compounds, seafood composition, and food processing. The octanol–water partition coefficient is an important indicator describing the tendency of a contaminant to adsorb to living organisms. In fact, benzophenone 3 and AHTN have high octanol–water partition coefficients, 3.79 and 5.7 [7], respectively, which indicate their high potential for bioaccumulation in aquatic organisms. Moreover, the fat content of seafood is another recognizable reason for the accumulation of contaminants [8]. Generally, higher levels of hydrophobic contaminants tend to accumulate and persist in seafood containing a high ratio of fat. 

Seafood is usually cooked to enrich taste, soften texture, increase safety, and improve nutrients digestibility and bioaccessibility. However, food processing and cooking (stewing, grilling, boiling/steaming, or frying), generally overlooked by authorities, can affect the stability of contaminants [9,10,11,12,13], including PCP. For example, Barbosa et al. [9] reported the reduction of AHTN and other musks, such as galaxolide (HHCB), in steamed crabs, compared to raw seafood. In the same study, the BP1 UV-filter was found to be present at lower levels in steamed mussels and gilthead seabream, compared to raw seafood. In another study, Trabalón et al. [8] reported the advantage of cooking (steaming and grilling) in terms of the reduction of galaxolide musk in cod and mackerel. 

The use of natural food additives, such as tea polyphenols (TP), is considered a safe method of increasing the storage life of seafood products. To date, several studies have reported that TP contains effective antimicrobial and antioxidant agents—mainly catechins—which have been incorporated into dry-seasoned squid [14], fish balls [15], and edible fish coating [16] to prolong shelf life and preserve fish quality. The use of green tea marinades, before cooking, has also been reported as a mitigation strategy for contaminants in meat [17]. However, the impact of green tea marinades on PCP stability and in fish in vitro bioaccessibility is still currently unknown.

The impact of contaminants on human health depends of chemical features, such as the nature and amount of contaminants in food, as well as on the amount of contaminants ingested that will be available for absorption in the gut after digestion. In general, only part of xenobiotics will become bioaccessible in the gastrointestinal tract. Thus, to provide a realistic risk of human exposure, food cooking procedures and bioaccessibility, generally overlooked, should be considered [18]. To the best of our knowledge, only Trabalón and collaborators [10] evaluated the bioaccessibility of PCP, namely galaxolide, in cod and mackerel. 

Over the years, to overcome the drawbacks inherent to in vivo (human or animal) assays, several in vitro bioaccessibility models (static or dynamic) have been developed to elucidate the food digestion process and estimate nutrient and toxin bioaccessibility (e.g., Cardoso et al. [19]). Static in vitro bioaccessibility models are cost- and time-effective, reproducible, and easy to implement compared to other models; thus they are often applied to assess the bioaccessibility of contaminants [10,11,12,13,20,21]. These models simulate the physiological conditions of the upper gastrointestinal tract (oral, gastric, and duodenal phases). 

The main goal of this study was to evaluate the impact of green tea marinade combined with different home-cooking procedures on the stability of AHTN and BP3, as well as the effect of in vitro digestion on these chemical hazards, using European seabass (*Dicentrarchus labrax*). For this purpose, seabass burgers were cooked in two different ways (fried and roasted), either directly or after marinating in green tea (*Camellia sinensis*). The outcomes achieved herein are of great value in estimating the amounts of AHTN and BP3 to which consumers may be exposed after ingestion of contaminated seafood, to ascertain the real impact on human health.

## 2. Results

### 2.1. Domestic Cooking 

The impact of cooking in AHTN and BP3 levels of European seabass are shown in Figure 1. The results are presented in dw (columns) as well as in relative percentage of AHNT and BP3 found following each cooking procedure (lines, raw levels = 100%). Analyses were performed in duplicate on each of the three burgers from each test group (i.e., control, AHTN, BP3). No statistically significant differences (*p* > 0.05) were verified between individual burgers. In general, cooking led to an overall reduction of AHTN, by about 45% (Figure 1). Cooking procedures (frying and roasted) did not present significant statistical differences (*p* > 0.05, Figure 1), despite a slight reduction in AHTN levels found in roasted compared to fried burgers. Marination promoted a statistically significant decrease in contaminant level (*p* < 0.05), with the average reduction ranging from 41% (raw) to 66% (roasted and frying). Concerning BP3, thermal processing also showed a 25% reduction in the contaminant level, regardless of the cooking procedure. The green tea marinade also diminished BP3 levels, with an average reduction of 26% when fried, 32% when roasted, and 37% raw.

### 2.2. Bioaccessibility

The bioaccessibility of AHTN and BP3 from raw, fried, and roasted European seabass, previously marinated, or not, with green tea, are summarized in Figure 2. The bioaccessibility of AHTN in cooked European seabass was considerably higher than in raw samples, ranging on average from 45% to 91%. Green tea marinade increased the bioaccessibility of AHTN, in particular when seabass was fried or roasted. Concerning BP3, low bioaccessibility values were found, in both raw and cooked samples, ranging on average from 6.3 to 19.3%. Green tea marination of European seabass burgers revealed increased bioaccessibility of BP3, especially when fried. Following absorption, BP3 can be hydroxylated and generate by-products, such as BP-1, BP-8, or THB [3], which could explain the low-mass balance percentages (<36%) obtained for BP3 (Table 1).

## 3. Discussion

### 3.1. Domestic Cooking 

Fish is usually consumed after being subjected to a culinary process that improves taste and texture [18]. Studies focusing on the influence of different cooking methods on seafood contaminant levels are of the utmost importance in order to achieve an accurate knowledge of intake, and, consequently, accurate risk assessment. This study involved six sample groups for each contaminant, namely raw, fried, and roasted European seabass, with or without green tea marination. European seabass was selected due to its high consumption (111,498 tonnes in 2018 in the European Union) [22] and aquaculture production in Europe. The process of frying the fish, in which the fish is immersed in oil that acts as a heat transferring compound, was chosen due to its high popularity around the world. Roasting fish in the oven is also a very typical culinary practice, and consumers usually have the perception that is a healthier cooking process. In both procedures, thermal destruction of microorganisms and enzymes occur, and a reduction of water content and activity on the surface of the food item is observed [23]. Additionally, during cooking, the denaturation of muscle fiber proteins results in tissues shrinking, leading to the release of fluids, including fat and water [24]. This fat loss can lead to the leakage of lipophilic compounds, such as AHTN and BP3. The reduction of contamination verified herein, using roasting and frying procedures, is consistent with a previous study reported by Barbosa et al., in which steaming promoted a loss of AHTN and other musks, such as galaxolide (HHCB), in crabs [9]. The same study verified the decrease of some UV-filters, including BP1, during the steaming of mussels and gilthead seabream (*Sparus aurata*). Trabalón et al. also showed a reduction of more than 30% in HHCB levels during the steaming and grilling of cod and mackerel [10]. Thermal cooking procedures have also been found to be effective in the degradation of various organic contaminants, including pesticide residues, in fish [9,25]

Green tea is particularly rich in (−)-epicatechin (EC), (−)-epigallocatechin (EGC), (−)-EC gallate (ECG), and (−)-EGC gallate (EGCG) [26]. These compounds have the ability to scavenge free radicals, and thus inhibit any free radical reactions [27]. This capacity is one of the explanations for the reduction of some contaminants, such as polycyclic aromatic hydrocarbons and heterocyclic amines, in meat marinated in green tea [27,28]. In fish, extracts rich in polyphenols can be used as active edible coatings in order to increase water-holding capacity and inhibit the survival of spoilage bacteria, therefore prolonging product shelf life [29]. Additionally, green tea has been shown to induce a reduction in lipid peroxidation in salmon fillets [30], while the ingestion of green tea has been shown to reduce the bioaccessibility of mercury in fish [21]. However, to the best of authors knowledge, the reduction of personal care contaminants, or other organic contaminants, through marination has not been reported. 

The composition of the marinating bath, the ratio of fish to liquid, and the exact method of treating the product are usually decisive in the quality of final product. Topus 2016 [31] demonstrated that salt and acetic acid concentration in the marinating process affected the physicochemical and sensory properties of marinated little tunny. The type of salt (NaCl or KCl) was also recently evaluated as a possible parameter that may influence contaminant levels in processed seafood [32]; the use of KCl instead of NaCl in smoked salmon processing resulted in lower levels of both polybrominated diphenyl ethers and polycyclic aromatic hydrocarbons. Despite these results, in this study salt was not considered, as its use is common in home-cooking procedures. 

Notwithstanding, the manifest decrease of AHTN and BP3 contamination, by the use of both green tea marinade and thermal cooking, does not eliminate the possibility of the subsequent generation of toxic metabolites, which should not be neglected. This study was only based on a targeted analytical approach, so further non-targeted evaluation should be considered in the future.

### 3.2. Bioaccessibility 

In general, raw and cooked samples revealed high protein digestibility (>90.3%, see Appendix A), indicating that the in vitro digestion protocol used in the present study successfully hydrolyzed and released almost all proteins to the bioaccessible fraction, thus revealing the efficiency of the in vitro digestive process. The highest protein bioaccessibility values registered after cooking were similar to those observed by Tavares et al. [33] in cooked hairtail fillets (frying and roasting). The authors reported a significant impact of frying on the content of released free amino groups, with significantly higher values in fried samples compared to raw samples. Frying grass carp fillets has also been reported to increase the content of released amino groups [34], and this increase was attributed to the hydrolysis of soluble proteins. The presence of proteins in fish, and the high levels of polyphenols in green tea marinade, can result in the formation of protein–polyphenol complexes, which can grow in size and even form sediments [35]. These complexes may also improve the stability of food emulsions on the shelf, prevent degradation of polyphenols, and improve target delivery to the intestinal tract [35].

This is the first study on the bioaccessibility of AHTN in seafood; thus, any comparison of the obtained results with the literature was not possible. Nevertheless, owing to the structural resemblance to HHCB, the high values obtained were in accordance with the results published by Trabalón et al. [10] from experiments with steamed and grilled cod and mackerel. Among culinary treatments, frying presents significant lower bioaccessibility compared to roasting. The high lipid content that occurs following the frying process may decrease the intestinal uptake of organic contaminants, since lipids are difficult to digest and can retain lipophilic contaminants, as previous reported by Cunha et al. [20] and Trabalón et al. [10].

Some benzophenones can still be present in the bioaccessible fraction, although not detected by the performed GC–MS analysis. Hence, a non-targeted follow-up study should be conducted to determine if digestion decreases BP3 bioaccessibility or generates even more toxic substances than the precursor, are if they are available to be adsorbed.

## 4. Materials and Methods

### 4.1. Standards and Chemicals

Tonalide (AHTN 98% purity) and 2-Hydroxy-4-methoxybenzophenone (BP3; 98% purity) were purchased from Promochem Iberia (Barcelona, Spain) and Alfa Aesar (Heysham, Lancashire, UK), respectively. The internal standards tonalide-d3 (AHTN-d3, 99% purity) and benzophenone-d10 (BPd10, purity > 98%) were acquired from Dr. Ehrenstorfer GmbH (Augsburg, Germany) and Sigma-Aldrich (Steinheim, Germany), respectively. The individual standards were prepared in acetonitrile at 1 g/L and stored at 4 °C. Stock solutions of each internal standard, at 2.5 µg/L in acetonitrile, were used as surrogate standards. 

HPLC-grade acetonitrile, used for extraction, was obtained from Honeywell, Riedel-de-Haën (Seetze, Germany). Trichloroethylene was purchased from Merck (Fontenay-sous-Bois, France). Deionized water, 0.055 µS/cm, was obtained with a Seralpur Pro 90CN from Seral (Ransbach-Baumbach, Germany). Sodium chloride was obtained from PanReac Quimica (Barcelona, Spain), and magnesium sulfate was acquired from Sigma-Aldrich (Japan). dSPE Enhanced Matrix Removal–Lipid (EMR-Lipid) was purchased from Agilent Technologies (Santa Clara, CA, USA). The reagents used to prepare the four digestion juices (saliva, gastric juice, duodenal juice, and bile) were the following: sodium bicarbonate (NaHCO_3_, Merck, ≥99.5%), calcium chloride dehydrate (CaCl_2_·2H_2_O, Sigma, ≥99%), potassium chloride (KCl, Merck, ≥99.5%), potassium thiocyanate (KSCN, Sigma, ≥99%), sodium dihydrogen phosphate (NaH_2_PO_4_, Merck, ≥99.5%), sodium sulfate (Na_2_SO_4_, Merck, ≥90%), sodium chloride (NaCl, Merck, ≥99.5%), ammonium chloride (NH_4_Cl, Riedel-de Haen, ≥99.5%), potassium dihydrogen phosphate (KH_2_PO_4_, Merck, ≥99.5%), magnesium chloride (MgCl_2_, Riedel-de Haen, ≥99.5%), hydrochloric acid (HCl, Merck, 37% w/w), urea (Sigma, 99–100.5%), glucose (Sigma, ≥99%), glucuronic acid (Sigma, ≥98%), D-(+)-glucosamine hydrochloride (Sigma, ≥99%), uric acid (Sigma, ≥99%), albumin from bovine serum (Sigma, pH7, ≥98%), α-amylase, from Aspergillus oryzae (Sigma, ~1.5 U/mg), mucin from the porcine stomach (Sigma, type II), pepsin from porcine stomach mucosa (Sigma, ≥400 units/mg protein), lipase from the porcine pancreas (Sigma, type II), pancreatin from the porcine pancreas (Sigma, meets USP testing specifications), trypsin from the porcine pancreas (Sigma, Proteomics Grade), α-chymotrypsin from the bovine pancreas (Sigma, type II), and bile porcine extract (Sigma).

### 4.2. Sampling and Sample Preparation

Preparation of fish burgers

Fillets (without skin or bones) from commercial-size European seabass from aquaculture production (N = 3, farmed, Dicentrarchus labrax) were acquired from a local market. The fillets were minced using a laboratory knife mill (Grindomix GM200, Retsch, Haan, Germany) and divided into three portions: (1) AHTN-spiked (400 µg/kg, on a wet weight basis, ww); (2) BP3-spiked (5000 µg/kg, ww); and (3) non-spiked (with equivalent volume of acetonitrile per gram of sample). Once prepared, samples were homogenized with a glass rod and stored at 4 °C for 1 h. Afterwards, six cylindrical burgers (10 g, 3.5 cm diameter, 1 cm height) were prepared from each portion of fresh minced fish, three of which were immersed in marinade (green tea marinade, see preparation below) for 1 h at 4 °C; the control group minced fish were left in the same conditions, without marination. 

The cooking conditions used were: (1) frying at 175 °C in 100 mL of sunflower vegetable oil (see composition in Appendix A) for 2 min (1 min each side) using a ^®^Teflon pan with a 1 L capacity, and (2) roasting at 200 °C for 10 min (5 min each side) using a Teflon^®^ pan. In both cooking procedures, basic parameters (i.e., time, temperature/potency) were controlled to achieve acceptable cooking (70 °C at the inner center of the burger). All individual burgers were weighed before and after cooking (Table 2). Moisture and lipid content were determined according to a standard method AOAC [36] (Table 2) in order to express results on a fresh and lipid basis. 

#### Preparation of Green Tea Marinade

Commercial green tea infusions (*Camellia sinensis*) were prepared from teas obtained from a local market (see chemical composition in Appendix A) by immersing three bags (1 g each) in 600 mL of boiling water for 3 min. After cooling, 100 mL of infusion was used to marinate three burgers.

### 4.3. In Vitro Digestion

The experimental procedure was based on a static digestion method described by Cunha et al. [20]. Initially, a composite sample of three burgers from each cooking method was prepared and minced with a scalpel to simulate mastication. In glass vials, 1.5 g of each sample was digested, in triplicate, at 37 °C using a Rotary Tube Mixer with Disc (25 rpm; LSCI, Portugal) inside of an incubator (Genlab, UK). In vitro digestion included the following steps: oral phase (4 mL of saliva fluid for 5 min at pH 7.0 ± 0.2), gastric phase (8 mL of gastric fluid for 2 h at pH 2.0 ± 0.2), and intestinal phase (8 mL of duodenal fluid and 4 mL of bile fluid for 2 h at pH 7.0 ± 0.2). To prevent enzyme degradation/inhibition, each digestion fluid was prepared just before starting the digestion protocol, and the pH was adjusted immediately before the digestion step with NaOH (1 M) or HCl (1 M). Following digestion, the reaction tubes were placed on ice to stop the digestion process and centrifuged at 2750× *g* at 10 °C for 10 min to separate the bioaccessible fraction (BIO) from sample residues (non-bioaccessible fraction, NBIO). Negative controls containing the digestion fluids, without a sample, were performed. Samples were kept at −20 °C until analysis.

AHTN and BP3 were quantified in the sample both before digestion (BD) in the BIO and NBIO fractions.

AHTN and BP3 in the bioaccessible fraction (%) were calculated as the following ratio:(BIO × 100)/BD
where BIO is the AHTN and BP3 levels detected in the bioaccessible (BIO) fraction, and BD is the amount of AHTN and BP3 detected in the sample before digestion.

Mass balance was evaluated using the flowing equation:%MB = ((ng bioaccessible + ng non-bioaccessible)/ng in cooked sample) × 100

### 4.4. Sample Extraction

Sample extraction was performed as described in Petrarca et al. [37]. Briefly, homogenized fish (≈1 g) or NBIO fractions (≈250 mg) were weighed into amber-glass vials, and internal standards were added. The sample was then extracted with acetonitrile and inorganic salts (MgSO_4_ and NaCl), followed by clean-up with EMR-Lipid. Afterwards, 1 mL of organic extract and 70 μL of trichloroethylene were added to 3 mL of deionized water. The cloudy solution formed was vortexed and centrifuged (1690× *g*, 5 min), and 100 µL of the sediment phase was collected in 2 mL amber-glass vials with an insert of 200 µL. 

BIO fractions (5 mL) were accurately transferred into 20 mL amber-glass vials containing 100 μL of internal standards (the solvent was previously evaporated). Then, 850 μL of acetonitrile and 150 μL of trichloroethylene were added to the sample. The cloudy solution was centrifuged for 2 min at 1690× *g*, and 100 µL of the bottom phase was collected in a 2 mL amber-glass vial, containing a 100 μL insert, for subsequent analysis. Two replicates of each sample were analyzed. 

### 4.5. Digestion Efficiency

To assess the efficiency of in vitro digestion, the total protein content was measured using a FP-528 DSP LECO nitrogen analyzer (LECO, St. Joseph, MO, USA; limit of detection = 0.84 mg N), as detailed by Alves et al. [38]. Analyses were performed on raw and cooked samples, before digestion (BD), as well as on BIO and NBIO fractions. Protein digestibility was always above 90%, leading to the conclusion that the in vitro digestion method was effective.

### 4.6. Instrumental Analysis

Analyses were performed in a Hewlett Packard HP6890 gas chromatography system equipped with a PAL LSI autosampler and interfaced to an Agilent 5973 single quadrupole mass selective detector with electron ionization (EI) source (Agilent Technologies, Santa Clara, CA, USA). Injections were performed in pulsed splitless mode (pulse pressure of 40 psi for 0.85 min and purge flow of 100 mL·min^−1^), and separation was performed in a 30 m × 0.25 mm × 0.25 mm Zebron-XLB capillary column (Phenomenex, Torrance, CA, USA). ChemStation software was used for data processing. Chromatographic and detection specifications were reported elsewhere [37]. 

### 4.7. Quality Control/Quality Assurance

Blank samples (fish free of analytes and in vitro digestion fractions obtained from the same samples) were spiked and used to evaluate linearity, sensitivity, precision, and accuracy (Appendix A), according to EU guidelines [39]. Matrix-matched calibration, with seven calibration levels (25 to 500 µg/kg for AHTN and 150 to 5000 µg/kg for BP3 in fish, and 12.5 to 300 µg/kg for AHTN and BP3 in in vitro digestion fractions), was used. Detection and quantification limits were calculated using low-level points to achieve signal-to-noise ratios of 3 and 10, respectively.

### 4.8. Statistical Analyses

Initially, the normal distribution of residuals and the homogeneity of variances were evaluated through the Shapiro–Wilk test (sample size N < 50) and Levene’s test, respectively. One-way ANOVA, or the Kruskal–Wallis test, were applied, according to residual distribution. Whenever statistical significances were found with one-way ANOVA, Tukey’s or Dunnett’s T3 post hoc tests were applied for mean comparison, depending on equal variance assumption or not. When statistical significances was found with the Kruskal–Wallis test, Dunn’s post hoc test was applied for means comparison. All analyses were performed at a significance level of 0.05 using SPSS software, version 28.0 (IBM Corporation, New York, NY, USA).

## 5. Conclusions

Cooking procedures (roasting and frying) led to a reduction of AHTN and BP3 levels in samples of European seabass muscle. In general, roasting was more effective than frying in reducing AHTN and BP3 levels in seabass. Green tea marination prior to cooking had a high impact on both ATN and BP3 reduction, when compared with non-marinaded seabass. Therefore, marinating fish with green tea prior to cooking could be a mitigation strategy to reduce personal care contaminant content in contaminated fish. Concerning bioaccessibility, a significant increase was observed in samples subjected to green tea marination, which was more evident in the case of AHTN. Overall, AHTN presented higher bioaccessibility (>45%) when compared with BP3 (<19%). Despite the great value of this study in assessing the exposure of consumers to AHTN and BP3 following seafood consumption, and thus contributing to the accurate evaluation of the impact on human health, new assays should be carried out to evaluate and identify possible hazard metabolites formed during cooking and ingestion.

## Figures and Tables

**Figure 1 molecules-27-04873-f001:**
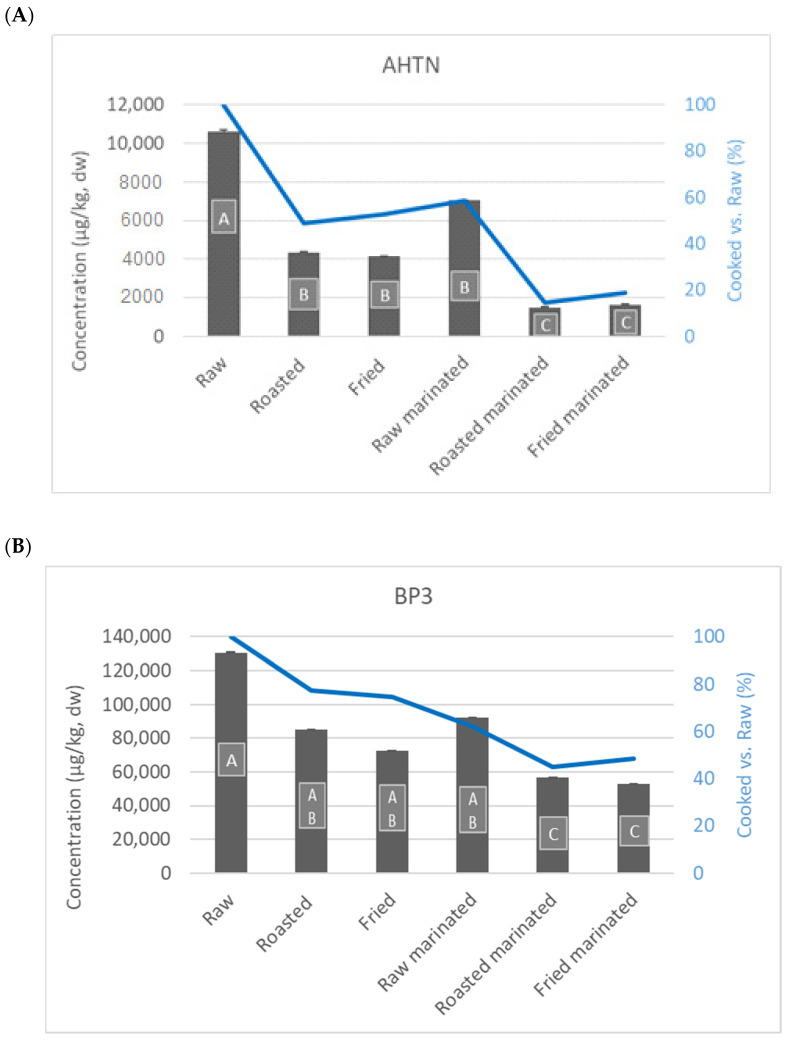
Impact of domestic cooking on AHTN (**A**) and BP3 (**B**) levels in European seabass (*Dicentrarchus labrax*). Different letters in each column show statistically significant differences (*p* < 0.05) from the given mean. Means were compared by Tukey HSD test or Dunnet’s T3 test, respectively, depending on whether or not homogeneity of variances was confirmed by Leven’s test.

**Figure 2 molecules-27-04873-f002:**
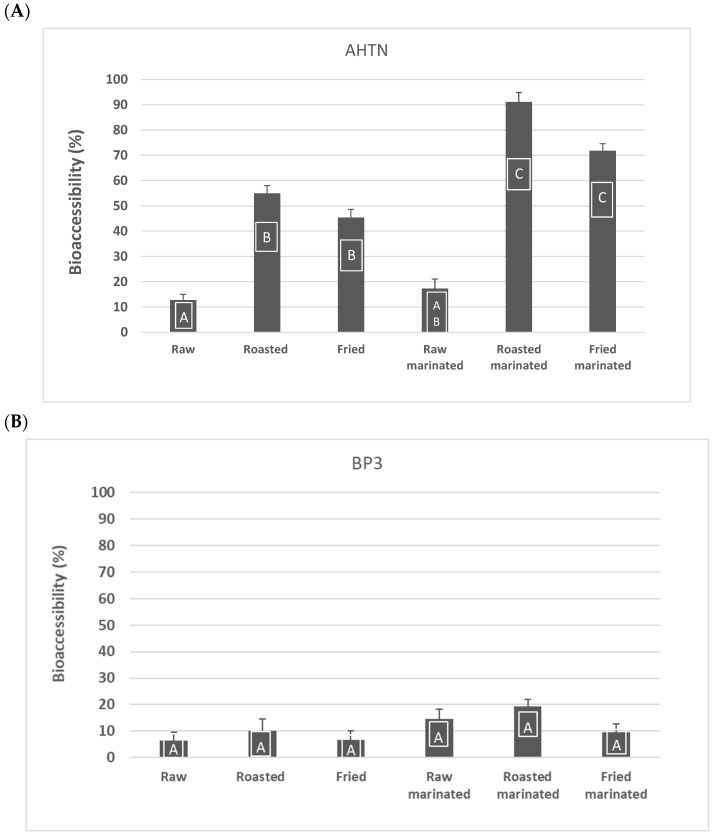
Bioaccessibility in the upper gastrointestinal tract of AHTN (**A**) and BP3 (**B**) from European seabass (*Dicentrarchus labrax*). Different letters in each column show statistically significant differences (*p* < 0.05) from the given mean. Means were compared by Tukey HSD test or Dunnet’s T3 test, respectively, depending on whether or not homogeneity of variances was confirmed by Leven’s test.

**Table 1 molecules-27-04873-t001:** Percentages (%MB) of the in vitro digestion experiment (mean ± standard deviation; *n* = 3).

Cooking Experiments	%MB *	
	AHTN	BP3
Roasted European seabass	98 ± 2 ^a^	35 ± 5 ^a^
Frying European seabass	98 ± 2 ^a^	36 ± 4 ^a^
Roasted European seabass previous marinade with green tea	98 ± 2 ^a^	32 ± 6 ^a^
Frying European seabass previous marinade with green tea	97 ± 3 ^a^	34 ± 5 ^a^

* MB (Mass balance) was calculated as MB = ((ng contaminant in bioaccessible fraction + ng contaminant in non bioaccessible fraction)/ng contaminant in cooked European seabass samples before digestion) × 100. Different letters in each column show statistically significant differences (*p* < 0.05) from the given mean.

**Table 2 molecules-27-04873-t002:** Weight loss after cooking and nutritional composition of raw and cooked seabass, with or without green tea marination.

	Weight Loss after Cooking (%)		Moisture (%)	
	*European seabass*	*Seabass marinade*	*European seabass*	*Seabass marinade*
Raw	-	-	65.77 ± 0.74 ^a^	69.80 ± 0.45 ^a^
Fried	18.69 ± 1.50 ^a^	21.69 ± 0.68 ^a^	53.83 ± 0.28 ^b^	58.79 ± 0.39 ^b^
Roasted	15.48 ± 0.98 ^a^	20.01 ± 0.58 ^a^	59.34 ± 2.44 ^b^	64.28 ± 0.91 ^a^
	**Fat Content (%)**		**Protein Content (%)**	
	*European seabass*	*Seabass marinated*	*European seabass*	*Seabass marinade*
Raw	22.51 ± 2.20 ^a^	22.48 ± 3.8 ^a^	20.21 ± 0.64 ^a^	19.87 ± 0.67 ^a^
Fried	25.92 ± 3.48 ^a^	25.76 ± 0.52 ^a^	25.89 ± 1.74 ^b^	22.61 ± 0.87 ^b^
Roasted	24.49 ± 1.27 ^a^	23.34 ± 1.60 ^a^	25.14 ± 0.54 ^b^	22.36 ± 0.06 ^b^

For each parameter, different letters in the same column show significant differences (*p* < 0.05).

## Data Availability

Not applicable.

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
