# Peer review of "Effects of Green Tea Marinade in the Bioaccessibility of Tonalide and Benzophenone 3 in Cooked European Seabass"

_molecules, 2022, doi:10.3390/molecules27154873_

Round 1
Reviewer 1 Report
The authors have investigated the impact of green tea marinated combined with different cooking home procedures on the stability and bioaccessibility of AHTN and BP3. The experiments have good practical application for food safety. However, the following aspects need further improvement before resubmission.
1. Only two figures and one table are presented in the results section to support the authors' findings. In addition, it is difficult for the reader to read and understand the meaning of the two figures drawn by the authors. For example, what do the A, B, C and AB in figures mean?
2. Both figures and one table do not meet to the requirement for a scientific paper.
3. Lines 121-149 should be placed in the introduction section.
4. The discussion section is not well written. The author did not compare their findings with the results of other researchers.
5. In lines 196-200, the chemical formula does not meet the requirement for a scientific paper.
Author Response
Dear Editor,
We appreciate your comments and those of the reviewers on our above mentioned manuscript. All of these comments were very helpful for revising and improving our paper, which has been revised in accordance with the recommendations. All the changes in the revised manuscript are highlighted and a detailed point-by-point response to each comment raised in the review process is provided below.
----------------------
The authors have investigated the impact of green tea marinated combined with different cooking home procedures on the stability and bioaccessibility of AHTN and BP3. The experiments have good practical application for food safety. However, the following aspects need further improvement before resubmission.
- Only two figures and one table are presented in the results section to support the authors' findings. In addition, it is difficult for the reader to read and understand the meaning of the two figures drawn by the authors. For example, what do the A, B, C and AB in figures mean?
Most of the tables and data are presented in supplementary material, we can add to the manuscript if you considered necessary. Different letters in each column of the graphic show statistically significant differences (p < 0.05) from the given mean.
- Both figures and one table do not meet to the requirement for a scientific paper.
The figures and tables were improved as requested.
- Lines 121-149 should be placed in the introduction section.
The paragraph describes previous research directly related to the sample cooking process proposed, so we prefer not to move it to the Introduction, because in this manner the reader can become aware of effects of cooking in the proximate composition. No modification has been made to the text.
- The discussion section is not well written. The author did not compare their findings with the results of other researchers.
А more detailed comparison is provided as requested, but we should mentioned that studies in musks and UV-filters are scarce.
- In lines 196-200, the chemical formula does not meet the requirement for a scientific paper.
As requested the chemical formula were written.
Reviewer 2 Report
Review molecules-1805941
Some related reports were missed in the section of introduction. For instance, tea components as coating materials on fish fillet quality and safety (For instance, Food and Bioprocess Technology, 10, 89-102).
Some related work on seabass quality attributes can be used for comparison and discussion, for instance, Food Chemistry, 354, 129581.
What were the effective components in the tea contributed the results? Further discussion is needed, incorporating more of the previous reports in this field. For instance, Food Hydrocolloids, 114, 106562; LWT-Food Science and Technology, 149, 111999.
Tables 1 and 2: significantly different analysis should be conducted and labeled to make the results more meaningful.
It is strange to put the Table 2 first and then Table 1. What was the justification?
The discussion should not be limited with detailed marinade but the common mechanism behind. For instance, previous work on reducing salt during marinade was reported (Food and Chemical Toxicology, 153, 112262), the mechanism may have positive meanings on the distribution of tea marinade, which should be discussed in-depth.
The integration of the results from different parameters should be enhanced.
Author Response
Some related reports were missed in the section of introduction. For instance, tea components as coating materials on fish fillet quality and safety (For instance, Food and Bioprocess Technology, 10, 89-102).
The reference was added to introduction as requested.
Some related work on seabass quality attributes can be used for comparison and discussion, for instance, Food Chemistry, 354, 129581.
The reference was added to the manuscript as requested.
What were the effective components in the tea contributed the results? Further discussion is needed, incorporating more of the previous reports in this field. For instance, Food Hydrocolloids, 114, 106562; LWT-Food Science and Technology, 149, 111999.
The information about the possible components in the tea that contribute to the results was added as requested.
Tables 1 and 2: significantly different analysis should be conducted and labeled to make the results more meaningful.
The information was added as requested.
It is strange to put the Table 2 first and then Table 1. What was the justification?
It was а mistake, the sequence of the tables was changed.
The discussion should not be limited with detailed marinade but the common mechanism behind. For instance, previous work on reducing salt during marinade was reported (Food and Chemical Toxicology, 153, 112262), the mechanism may have positive meanings on the distribution of tea marinade, which should be discussed in-depth.
The discussion of marinated was improved as requested.
The integration of the results from different parameters should be enhanced.
The revision was made as requested.
Round 2
Reviewer 2 Report
The authors have addressed the questions quite well. There are no further comments.